# Genomic Survey and Resources for the Boring Giant Clam *Tridacna crocea*

**DOI:** 10.3390/genes13050903

**Published:** 2022-05-18

**Authors:** Juan Antonio Baeza, Mei Lin Neo, Danwei Huang

**Affiliations:** 1Department of Biological Sciences, Clemson University, Clemson, SC 29634, USA; 2Departamento de Biología Marina, Universidad Catolica del Norte, Coquimbo 1780000, Chile; 3Smithsonian Marine Station at Fort Pierce, Smithsonian Institution, Fort Pierce, FL 34949, USA; 4Department of Biological Sciences, National University of Singapore, Singapore 117558, Singapore; tmsnml@nus.edu.sg (M.L.N.); huangdanwei@nus.edu.sg (D.H.); 5Tropical Marine Science Institute, National University of Singapore, Singapore 119227, Singapore; 6Lee Kong Chian Natural History Museum, National University of Singapore, Singapore 117377, Singapore

**Keywords:** genome skimming, low-pass genome sequencing, genomic resources

## Abstract

The boring giant clam *Tridacna crocea* is an evolutionary, ecologically, economically, and culturally important reef-dwelling bivalve targeted by a profitable ornamental fishery in the Indo-Pacific Ocean. In this study, we developed genomic resources for *T. crocea*. Using low-pass (=low-coverage, ~6×) short read sequencing, this study, for the first time, estimated the genome size, unique genome content, and nuclear repetitive elements, including the 45S rRNA DNA operon, in *T. crocea*. Furthermore, we tested if the mitochondrial genome can be assembled from RNA sequencing data. The haploid genome size estimated using a k-mer strategy was 1.31–1.39 Gbp, which is well within the range reported before for other members of the family Cardiidae. Unique genome content estimates using different k-mers indicated that nearly a third and probably at least 50% of the genome of *T. crocea* was composed of repetitive elements. A large portion of repetitive sequences could not be assigned to known repeat element families. Taking into consideration only annotated repetitive elements, the most common were classified as Satellite DNA which were more common than Class I-LINE and Class I-LTR Ty3-gypsy retrotransposon elements. The nuclear ribosomal operon in *T. crocea* was partially assembled into two contigs, one encoding the complete ssrDNA and 5.8S rDNA unit and a second comprising a partial lsrDNA. A nearly complete mitochondrial genome (92%) was assembled from RNA-seq. These newly developed genomic resources are highly relevant for improving our understanding of the biology of *T. crocea* and for the development of conservation plans and the fisheries management of this iconic reef-dwelling invertebrate.

## 1. Introduction

Bivalves (class Bivalvia) belong to a species-rich monophyletic clade of molluscs (phylum Mollusca) with remarkable morphological, physiological, behavioral, and ecological disparity [1]. Among them, the giant clams (fam. Cardiidae, subfam. Tridacninae) are renowned because of their obligate mutualism with photosynthetic dinoflagellates (fam. Symbiodiniaceae) [2,3] and a brightly colored hypertrophied mantle that typically projects beyond the valves and is exposed to light most of the time [4]. Iridocytes residing within the mantle disperse productive wavelength to the photosynthetic dinoflagellates harbored (extracellularly) within a tubular network derived from its digestive system [5,6]. All recognized species of giant clams have traditionally been targeted by subsistence and commercial fisheries across most of their range of distribution [7,8], and more recently also threatened by anthropogenic activities such as pollution [9] and global warming [10]. Ongoing giant clam aquaculture efforts have helped to restock over-exploited wild populations [11,12], and meet commercial demands for the profitable ornamental aquarium trade [13].

Among species of *Tridacna*, the boring giant clam *T. crocea* is the smallest, reaching up to 15 cm in shell length. *Tridacna crocea* species exhibits a remarkable lifestyle, where it spends its entire benthic life completely entrenched in coral boulders and reef rocks in contrast to the archetypal epibenthic lifestyle exhibited by all other giant clams [14] (Figure 1a). Boring of coral rock by *T. crocea* is accomplished through acid secretion by pedal mantle tissue [15]. The boring giant clam is an important prey to a wide array of predators and scavengers, as well as a key contributor to reef bioerosion [16,17]. *Tridacna crocea* has a geographic range from the Eastern Indian Ocean, across the Indo-Malay Archipelago, to the Western Pacific [7,18]. Over the past three decades, *T. crocea* has been over-exploited across most of its range of distribution and data suggest that wild populations are dwindling at an alarming rate in certain regions [7]. The same holds true for most if not all other species in the genus [7]. As a result of intense exploitation, *T. crocea* and all other congeners are listed in the Appendix II of the Convention of International Trade in Endangered Species (CITES) [19]. Despite the evolutionary significance, ecological relevance, and cultural and commercial value of *T. crocea*, only a few but increasing number of genetic (e.g., [20,21]) and genomic (e.g., [22,23]) resources exist for this species. The development of additional genomic resources for the boring giant clam *T. crocea* is most important; they can help us to improve our understanding of the biology of this species and boost the development of conservation plans and the fishery management of this iconic reef-dwelling invertebrate.

This study forms part of a broad effort aimed at developing genomic resources in *T. crocea* and other giant clams. Using a low-pass (~6×) short read next generation sequencing (NGS) strategy, this study (i.) estimated the nuclear genome size using an in silico k-mer approach, (ii.) determined unique genome content (single- and low-copy genome elements), (iii.) discovered, annotated, and characterized nuclear repetitive elements, (iv.) assembled and annotated the 45S rRNA DNA operon, and (v.) tested if mitochondrial genomes can be assembled from RNA-seq. The newly developed resources are of utmost relevance to continue improving our understanding of the biology, conservation, and fisheries management of this widespread giant clam species.

## 2. Material and Methods

### 2.1. Sampling and DNA Extraction

One specimen of *T. crocea*, whose origin was Palau, was purchased from a local aquarium dealer (Fresh N Marine, Singapore, Singapore). Permits for specimen collection, export, and import (e.g., CITES) from Palau were obtained by the aquarium store. Total genomic DNA was extracted from adductor muscle tissue (to avoid sequencing symbiotic dinoflagellates) of the captured specimen using the DNeasy Blood and Tissue Kit (Qiagen, Singapore) following the manufacturer’s protocol. Extracted DNA was then transported to the Department of Biological Sciences, National University of Singapore, Singapore for library preparation.

### 2.2. Library Preparation and Sequencing

An Illumina paired-end shotgun library was prepared using the NEBNext Ultra DNA Library Prep Kit for Illumina (New England BioLabs, Singapore, Singapore) following the standard manufacturer’s protocol and sequenced in an Illumina MiSeq system (Illumina, San Diego, CA, USA) using a 2 × 250 cycle. A total of 17,397,666 pairs of (PE) reads were generated and deposited in the short read archive (SRA) repository (Bioproject: PRJNA607047; BioSample: SAMN14120369; SRA accession number: SRS6236858) at GenBank. This dataset was generated by [23] to assemble the mitochondrial genome of *T. crocea* and examine phylogenetic relationships among representatives of the subfamily Tridacninae.

### 2.3. Genome Size Estimation in Tridacna crocea

Low-quality sequences (Phred scores < 20) and Illumina adapters were removed using the software fastp v.0.20.1 [24] with default parameters, leaving 17,244,752 high-quality PE reads. Next, contaminants (bacteria, archaea, virus, and human reads) were removed using the taxonomic classification system Kraken2 v2.1.2 [25] with the pre-built database kraken2-microbial-fatfree (https://lomanlab.github.io/mockcommunity/mc_databases.html) (accessed on 15 January 2022), leaving 16,931,914 contaminant-free high-quality PE reads. These clean and filtered PE reads were used for the estimation of genome size by counting k-mers of different word sizes (21, 24, 27, 30, 33) in the software KMC 3 v. 3.2.1 [26]. The k-mer frequency distribution was then processed with the program REPeat SPECTra Estimation (RESPECT) v.1.0.0 [27].

### 2.4. Repetitive Elements in the Nuclear Genome of Tridacna crocea

Discovery, annotation, and quantification of repetitive elements in the genome of *T. crocea* were conducted as described in [28,29] using the pipeline RepeatExplorer v.2.3.8 [30,31] available in the platform Galaxy (http://repeatexplorer.org/—accessed on 15 January 2022) [32]. RepeatExplorer was developed for the efficient analysis of repetitive element’s composition and abundance in plant and animal genomes using low-pass short read Illumina PE sequences [30,31]. After an initial clustering of the reads, RepeatExplorer assembled contigs using the software CAP3 [33] and annotated these assembled contigs using the Metazoa v.3.0 repeat dataset that is included in the pipeline. All other parameters in RepeatExplorer were set to default values. The genome proportion of each identified repetitive element cluster was calculated as the percentage of reads [30].

RepeatExplorer is most efficient in discovering repetitive elements when short reads do not overlap [31]. We assessed the extent of overlap in the analyzed PE reads using the script ‘Scan paired-end reads for overlap’ in RepeatExplorer before running the analysis and found that a relatively large proportion (88%) of the analyzed reads did overlap. Therefore, we discarded overlapping reads and ran the pipeline using those reads that did not overlap (*n* = 3,780,214 PE reads). Furthermore, the first RepeatExplorer analysis failed to retrieve the nuclear ribosomal operon (45S rRNA DNA operon) of *T. crocea*. A preliminary assembly of the reads using the software SPAdes 3.11 [34] Bankevich et al., 2012 as implemented in the program Shovill (https://github.com/tseemann/shovill—accessed on 15 January 2022), classified as contaminants by Kraken2 retrieved contigs that matched to the 18S and 28S molluscan nuclear ribosomal genes (part of the 45S rRNA DNA operon) after BLAST-searching assembled contaminant contigs against the non-redundant (nr) nucleotide NCBI database (with *E*-values < 1 × 10^–6^). We inferred that Kraken2 filtered out reads belonging to the 45S rRNA DNA operon of *T. crocea*, likely due to the similarity of this repetitive element among eukaryotes, including gastropods and humans. Thus, we ran a second RepeatExplorer analysis using non-overlapping reads (*n* = 4,228,670 PE reads) obtained prior to contaminants (bacteria, archaea, virus, and human reads) removal by Kraken2 to determine the relative abundance of the nuclear ribosomal operon in the genome of *T. crocea*. We considered that the low number of reads flagged as contaminants by Kraken2 (<1%, see above) likely did not impact (at least considerably) these second quantification of repetitive elements in the genome of *T. crocea* (also, see [28,29]).

### 2.5. Nuclear Ribosomal Operon in Tridacna crocea

The nuclear ribosomal operon, that codes for the large and small nuclear rRNA genes (18S or ssrDNA, 28S or lsrDNA), the 5.8S rDNA gene, two internal transcribed spacers (ITS1 and ITS2), and two external transcribed spacers (5′ ETS and 3′ ETS), in the genome of *T. crocea* was retrieved using a strategy modified from [35] that included inspecting the contigs assembled by the program CAP3 annotated as the 45S rRNA DNA by RepeatExplorer. Each retrieved contig was then blasted against the non-redundant (nr) nucleotide NCBI database as well as Dfam [36] and Rfam [37] and aligned using the script number in the program MUMmer4 v4.0.0rc1 [38] to the complete nuclear ribosomal operons of *Perna canaliculus* and *P. viridis* available in Genbank (accession number: MK419107 and MK419106, respectively—[39]). If the contigs did match bivalve ribosomal sequences (with *E*-values < 1e^−6^) and mapped to the nuclear ribosomal operons belonging to *Perna* spp., then the exact coding positions of the 18S and 28S nuclear rDNAs and the boundaries of the 5′ and 3′ ETSs were determined using the program Barrnap (https://github.com/tseemann/barrnap—accessed on 15 January 2022), with the eukaryote database. The exact coding positions of the 5.8S nuclear rDNA and the boundaries of the ITS1 and ITS2 were determined using ITSx v. 1.1b1 [40].

### 2.6. Mitochondrial Genome Assembly from RNA-seq in Tridacna crocea

The raw sequence data used to assemble the mitochondrial genome of *T. crocea* from RNA reads were generated by [22] to understand the origin of photosymbiosis in marine bivalves and detailed information on sampling, DNA extraction, and sequencing methods can be found in the aforementioned paper. The RNA-seq dataset (37,565,862 PE reads) was downloaded from GenBank (SRA Accession number: SRR11252443), and the mitochondrial genome was assembled with the specialized pipeline MITGARD [41]. MITGARD first mapped the reads to a reference mtDNA provided by the user and *de novo* assembled the mapped reads using the programs Trinity 2.8.5 [42] and rnaSPAdes 3.13.1 [34,43]. In parallel, the pipeline assembled the genome using a genome-guided strategy (in the software Trinity), mixed the totality of the assembled contigs, mapped these assembled contigs to the reference mitochondrial genome, and finally converted the contigs into a mitochondrial genome assembly [41]. We used the complete mitochondrial genome of *T. crocea* (MT127796) retrieved from GenBank as the reference. The mitochondrial genome assembly used k-mer sizes of 33 and 49. Reads were quality trimmed with fastp prior to mitochondrial genome assembly using MITGARD. The complete or partially assembled mitochondrial genome were first annotated in silico using the web servers MITOS (http://mitos.bioinf.uni-leipzig.de) and MITOS2 (http://mitos2.bioinf.uni-leipzig.de—accessed on 15 January 2022) [44] with the invertebrate genetic code (code 5). A consensus between the two in silico annotations was achieved during manual curation of the aforementioned annotations, including start and stop codon error corrections. Genome visualization was performed in Chloroplot (https://irscope.shinyapps.io/Chloroplot—accessed on 15 January 2022) [45].

## 3. Results and Discussion

### 3.1. Genome Size Estimation in Tridacna crocea

The average haploid genome size of *T. crocea* estimated using a k-mer approach varied between 1,311,675,074 (1.31 Gbp, with kmer = 33) and 1,391,427,111 bp (1.39 Gbp, with kmer = 21). A slight decrease in genome size estimation was observed with concomitant increases in k-mer word size. In turn, estimated unique genome content (single and low-copy genomic elements) ranged from 33% (with kmer = 21) to 50% (with kmer = 31) in *T. crocea*. Unique genome content increased asymptotically with k-mer word size, indicating that nearly a third and probably at least 50% of the genome of *T. crocea* is composed of repetitive elements.

Genome size varies considerably in bivalves (class Bivalvia), from 0.64 Gbp in the oyster *Crassostrea gigas* (fam. Ostreidae) to 5.28 Gbp in the nutclam *Acila castrensisnotus* (fam. Nuculidae) (Animal Genome Size Database http://www.genomesize.com/—accessed on 15 January 2022 [46]) (Figure 1b). Most genome sizes in the aforementioned database are based on C-values estimated using flow cytometry, Feulgen image analysis densitometry, and bulk fluorometric assays (also, see [35]). Within the Cardiidae, the family to which *T. crocea* belongs, the genome size (GS) has been estimated only in four non-tridacnine species; *Cerastoderma edule* (GS = 1.34 Gbp), *C. pinnulatum* (GS = 1.37 Gbp), *Laevicardium mortoni* (GS = 1.27 Gbp), and *Trachycardium quadragenarium* (GS = 1.96 Gbp) [47,48]. The estimated genome size of *T. crocea* is well within the observed range in representatives of the Cardiidae as well as in bivalve molluscs. Importantly, the relatively large genome size (>1 Gbp) coupled with the relatively low unique genome content suggests that both short reads (i.e., Illumina, BGI) and long-reads (i.e., Pacific Biosciences and/or Oxford Nanopore Technology) together with chromosome conformation capture techniques (e.g., Hi-C) will likely be necessary to assemble a high-quality (i.e., chromosome-level and/or telomere-to-telomere) genome in *T. crocea*.

### 3.2. Repetitive Elements in the Nuclear Genome of Tridacna crocea

In the first analysis of repetitive elements in the genome of *T. crocea*, the pipeline RepeatExplorer automatically selected a sub-sample of 827,430 reads to be analyzed, of which a total of 416,924 reads were contained in 55,852 clusters. The proportion of reads comprising the top 289 clusters that represented the most abundant repetitive elements in the genome of *T. crocea* was relatively high (32%, *n* = 264,778 reads). Importantly, a large portion (88.2%) of the top repetitive element families (*n* = 255 clusters containing 242,117 reads) were reported as “unclassified” by RepeatExplorer; these read clusters could not be assigned to any known repeat family. Taking into consideration only clusters that were annotated in RepeatExplorer (*n* = 34), the most abundant repetitive elements were classified as Satellite DNA (*n* = 15 clusters, 9400 reads) which were more common than Class I—Long Interspersed Nuclear Element (LINE) (*n* = 9 clusters, 3939 reads) and Class I—Long Terminal Repeat (LTR) Ty3-gypsy retrotransposon (*n* = 2 cluster, 5164 reads) elements. Class II—Subclass 2 Maverick mobile elements (*n* = 5, 2437 reads) and unclassified Class I—LTR retrotransposons (*n* = 2, 2808 reads) were less common in the nuclear genome of *T. crocea*. No clusters were classified as 45 s rRNA DNA in this first analysis (Figure 2).

The second analysis of repetitive elements in RepeatExplorer resulted in a repeatome profile similar to that of the first analysis. In this second analysis, RepeatExplorer automatically selected a sub-sample of 791,674 reads to be analyzed, of which a total of 427,017 reads were contained in 69,908 clusters. The proportion of reads comprising the top 269 clusters was relatively high (31%, *n* = 245,419 reads). As observed in the first analysis, a large portion (93%) of the top repetitive element families (*n* = 246 clusters containing 227,976 reads) were reported as “unclassified” by RepeatExplorer. Annotated clusters (*n* = 23) included Satellite DNA (*n* = 9 clusters, 4214 reads), Class I—LINE (*n* = 5, 3020 reads) and Class I—LTR Ty3-gypsy retrotransposon (*n* = 3 cluster, 6560 reads), and unclassified Class I—LTR retrotransposons (*n* = 2, 2471 reads) plus the 45S rRNA DNA retrieved as two different clusters, one classified as 18S rRNA DNA (368 reads) and the second classified as 28S rRNA DNA (253 reads), both with relatively low abundance. In this second analysis, Class II—Subclass 2 Maverick mobile elements were not detected, and an additional single cluster classified as Class II—Subclass 1 TIR/Tc1 Mariner element (269 reads) was retrieved (Figure 2).

Our analysis of the ‘repeatome’ in *T. crocea* detected a considerably large number of repetitive elements that were not annotated, in agreement with previous studies exploring the ‘repeatome’ in other bivalves, molluscs, and invertebrates in general [49,50]. The above indicates that many new repetitive elements will be discovered by future studies focusing on the repetitive elements of *T. crocea*, other congeneric and co-familiar species, and bivalves in general.

Only recently, the ‘repeatome’ of bivalves has been exhaustively studied due to the increasing availability of sequenced genomes. These few studies have revealed that repetitive elements are quite abundant (e.g., 32.9% in *Ruditapes philippinarum* [1.12 Gb]; 33.7% in *Sinonovacula constricta* [1.34 Gbp]; 42.2% in *Archivesica marissinica* [Genome size = 1.52 Gbp]; 43.14% in *Cyclina sinensis* [0.9 Gbp]; among others) [49,50] in bivalve genomes. Furthermore, the expansion of repetitive elements is in great part a driver of genome size in bivalves [49]. A large number of non-annotated repetitive elements precludes at this time any comparison of these components of the genome landscape between *T. crocea* and other bivalve species with assembled and annotated nuclear genomes using large DNA-seq datasets.

### 3.3. Nuclear Ribosomal Operon in Tridacna crocea

The 45S rRNA DNA operon of *T. crocea* was partially reconstructed using the aforementioned approach. One contig assembled by the pipeline RepeatExplorer (2841 bp in length) was determined to encode the ssrDNA and the 5.8S rDNA gene after BLAST-searching it against the nr NCBI, Dfam, and Rfam databases. This contig did match bivalve ribosomal sequences belonging to the class Bivalvia available in GenBank with *E*-values < 1 × 10^−6^. This first contig was comprised of a partial 5′ ETS (length = 540 bp), the ssrDNA (1869 bp, full sequence [fs]), ITS1 (249 bp [fs]), a 5.8S rDNA (154 bp [fs]), and a partial ITS2 (156 bp). The second contig (matching bivalve ribosomal sequences with E-values < 1 × 10^−6^) was comprised of a partial lsrDNA (1767 bp, 60% complete as reported by the program Barrnap) and 3′ ETS (541 bp [likely partially assembled]) (Figure 3) (Appendix A).

Historically, the 45S rRNA DNA operon has been instrumental in revealing phylogenetic relationships at multiple taxonomic levels in invertebrates, including the phylum Mollusca [23,51,52]. Nonetheless, to the best of our knowledge, studies examining the structure and organization of the nuclear ribosomal RNA gene operon are almost non-existent in bivalves (see [39], for an exception). We argue that our strategy to assemble the 45S rRNA DNA operon in *T. crocea* can be replicated in other closely related species in order to study the phylogenetic relationships among representatives of the family Cardiidae (in addition to mitochondrial genomes—[23]). Furthermore, with larger sequence datasets, the 45S rRNA DNA operon might be assembled in its entirety (see [35]), setting the stage for a more detailed analysis of its structure. Although only partially assembled, the newly described ribosomal operon can enable the development of new gene-targeted phylogenomic approaches using low coverage sequencing (as an alternative or addition to, for example, ultraconserved elements [53] and anchored hybrid enrichment [54] to study genealogical relationships among representatives of the family Cardiidae and other bivalves.

### 3.4. Mitochondrial Genome Assembly from RNA-seq in Tridacna crocea

The program MITGARD assembled a nearly complete mitochondrial genome of *T. crocea* in 11 different contigs (range: 269–3569 bp in length) that comprised 92.06% of the reference mitochondrial genome (reference length: 18,266 bp) (Appendix A). The nearly complete mitochondrial genome assembly retrieved from RNA-seq was a close match to the reference genome (p-distance = 0.01838), exhibited an AT-skew (A = 4.500, T = 5.975, C = 2329, G = 4009), and encoded 12 protein-coding genes (PCGs, all retrieved complete), two ribosomal RNA genes (rrnS [12S ribosomal RNA] and rrnL [16S ribosomal RNA]), and 3 transfer RNA (tRNA) genes (trnS2, trnH, and trnI) (Figure 4). The PCG *nad2* as well as the tRNA genes trnA, trnC, trnD, trnE, trnF, trnG, trnK, trnL1, trnL2, trnM, trnN, trnP, trnQ, trnR, trnS1, trnT, trnV, trnW, and trnY, were missing. The gene order observed in this almost complete mitochondrial genome was identical to that reported recently for *T. crocea* [23].

Several studies have explored the population genetics in *T. crocea* across the Indo-West Pacific, focusing on specific regions such as the Indo-Malay Archipelago [20,21,55,56] or Indo-Australian Archipelago [57,58] using a short mitochondrial marker (*cox1*) but the extent of genetic/genomic connectivity among the many other populations of this widespread species (e.g., Southeast Asia and East Asia) is not currently known. We argue in favor of additional studies combining whole mitochondrial genomes either retrieved from DNA-seq [23] or RNA-seq (this study) coupled or not with nuclear markers (e.g., 45S rRNA DNA [see above] and SNPs) to assess population genomic structure and connectivity in the boring giant clam *T. crocea* across its entire range of distribution in the Indo-Pacific Ocean.

## 4. Conclusions

This study is among the first developing genomic resources for *T. crocea*, an evolutionarily, ecologically, economically, and culturally significant reef-dwelling invertebrate that is the target of a profitable ornamental fishery in the Indo-Pacific Ocean and experiencing major environmental problems. We have employed a series of bioinformatics tools that can retrieve relevant genomic information from low-pass short-read Illumina sequencing datasets. Using these tools, we have determined the genome size, we have estimated unique content, discovered, partially annotated, quantified nuclear repetitive elements, and assembled (partially) and annotated the ribosomal RNA operon. Furthermore, we tested and succeeded in assembling a nearly complete mitochondrial genome from RNA-seq data. This information is expected to guide a chromosome-level assembly of the nuclear genome of *T. crocea*, boost our understanding of the biology of this remarkable bivalve, and help explore the genomic underpinnings of the acclimatization and adaptation to local and global climate change in *T. crocea*.

## Figures and Tables

**Figure 1 genes-13-00903-f001:**
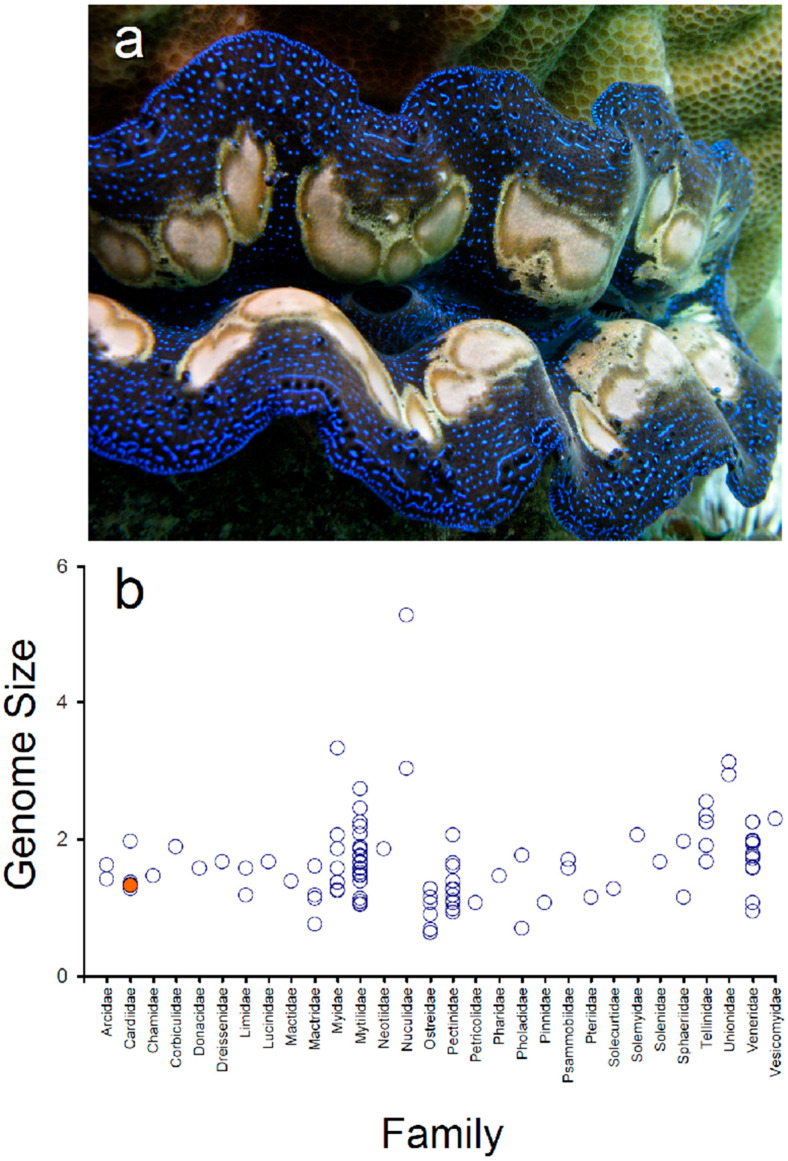
Genome size estimation (Gigabases) using a k-mer approach in the boring giant clam *Tridacna crocea* (orange dot) (**a**) and genome size estimated for other species belonging to different families in the class Bivalvia (open circles) (**b**). Genome size data from other bivalves retrieved from www.genomesize.com accessed on 15 January 2022. The top photograph depicts a specimen of *T. crocea* with a blue mantle color pattern (photo credit: J. Antonio Baeza).

**Figure 2 genes-13-00903-f002:**
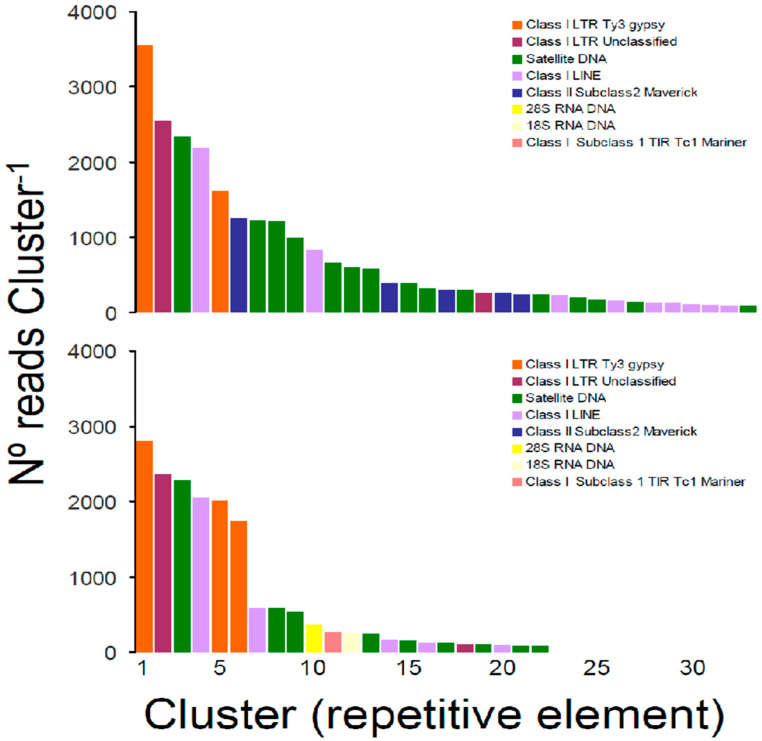
Size distribution and repeat composition of annotated clusters generated by similarity-based partitioning in the boring giant clam *Tridacna crocea*. Bars are colored according to the type of repeat present in the cluster, as determined by the similarity search in RepeatExplorer2. The first analysis (**top**) was conducted only with non-overlapping reads after contaminant filtering using the program Kraken2. The second analysis (**bottom**) was conducted only with non-overlapping reads but without contaminant filtering. See text for further details.

**Figure 3 genes-13-00903-f003:**
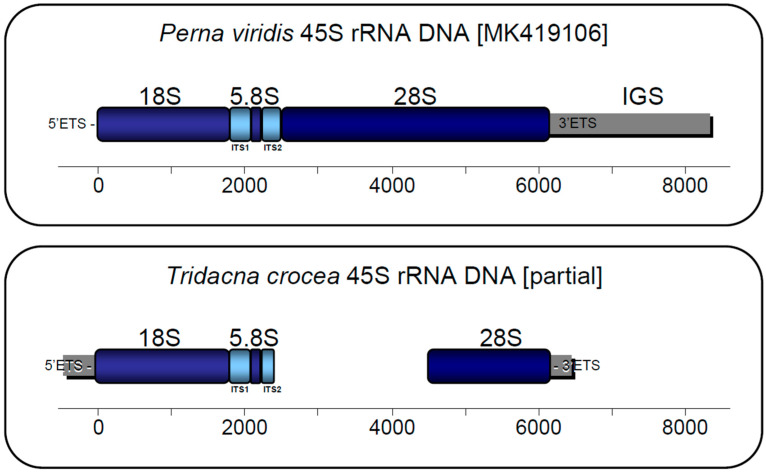
Partially assembled 45S rRNA DNA of *Tridacna crocea* and comparison with the complete 45S rRNA DNA of *Perna viridis* (GenBank accession number: MK419106.1).

**Figure 4 genes-13-00903-f004:**
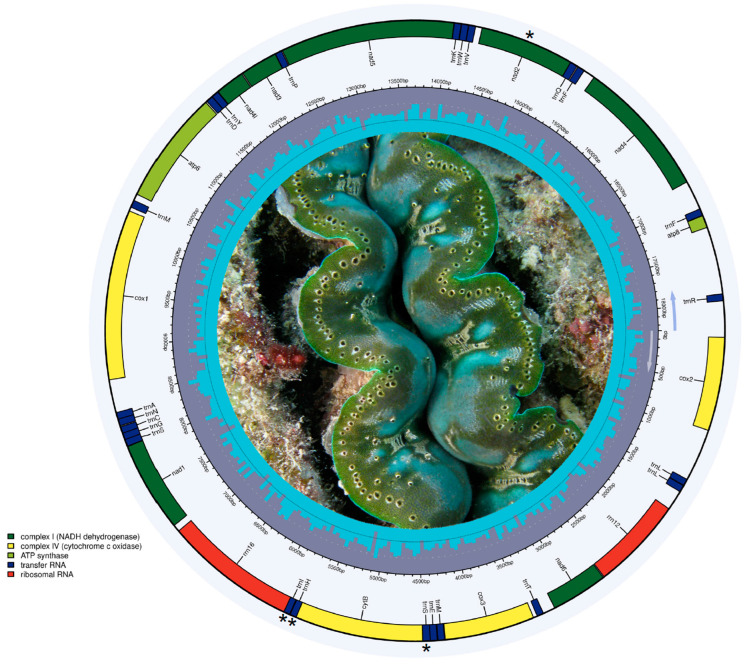
Circular genome depiction of the mitochondrial genome in *Tridacna crocea* based on RNA-seq data from [22]. Asterisks indicate transfer RNA genes retrieved and protein-coding genes not retrieved from RNA-seq data (photo credit: J. Antonio Baeza).

## Data Availability

The whole-genome sequencing data are available in the NCBI Sequence Read Archive (SRA) repository (Bioproject: PRJNA607047; BioSample: SAMN14120369; SRA accession number: SRS6236858). The RNA sequencing data are available in the NCBI SRA repository (SRR11252443).

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
