# Peer review of "Genomic Survey and Resources for the Boring Giant Clam Tridacna crocea"

_genes, 2022, doi:10.3390/genes13050903_

Round 1

Reviewer 1 Report

Baeza et al. carried out a genomic analysis of the boring giant clam Tridacna crocea, a species of bivalve in the family Cardiidae, to explore its genome size, unique genome content, repetitive elements, nuclear RNA gene operon, and mitochondrial chromosome assembly. This study provides the genomics resource of an endangered species which is listed in the Convention on International Trade in Endangered Species (CITES) appendix II (https://cites.org/eng), and the International Union for Conservation of Nature (IUCN) Red List of Threatened Species (Version 2017-2, www.iucnredlist.org).
I commend the authors for taking this initiative which is crucial for the scientific measures and subsequent policy making to preserve the flora and fauna of tropical and subtropical shallow sea coral reef environments.
I have the following comments/suggestions for the authors:
Introduction
1. What is low-pass short read next generation sequencing? Why have authors opted for this technique instead of deep sequencing?
2. What is the genome coverage authors have obtained using low-pass short read next generation sequencing? Mention it in the manuscript.
Methodology
1. Line 107-109: “This dataset was generated by [22] Tan et al. (2021) to assemble the mitochondrial genome of T. crocea and examine phylogenetic relationships among representatives of the same subfamily Tridacninae.” What is this dataset here?
2. During preprocessing of the data, what is the minimum read length selected by the authors? After quality filtering, what was the %age of paired-end and singleton reads? Have authors removed the singleton reads before the analysis? Explain it.
3. Authors have used two different genome-assembly strategies viz., SPAdes and CAP3, and found the discrepancies in the results. Rather, was it not useful to de
novo assemble the reads first using SPAdes followed by the contig assembly using CAP3?
4. Authors have used the Kraken2-microbial-fatfree database to filter out the microbial reads? Then, how is it possible that Kraken2 filtered out the 45 rRNA DNA operon genes? Have authors tried changing the parameters of Kraken2 before concluding such results?
5. Why have the authors aligned the nuclear ribosomal operons of T. crocea with those of Perna canaliculus and Perna virdis?
6. What is the “reference mtDNA provided by us”?
7. Authors have assembled the mitochondrial genome using de novo strategies using Trinity and rnaSPAdes. “In parallel, the pipeline assembled the genome using a genome-guided strategy (in the software Trinity), mixed the totality of the assembled contigs, mapped these assembled contigs to the reference mitochondrial genome, and finally converted the contigs into a mitochondrial genome assembly ([39] Nachtigall et al. 2021).”
8. What do you mean by “parallel” and “converted the contigs into a mitochondrial genome assembly ([39] Nachtigall et al. 2021).” What is this reference for? Authors must be cautious about the technical terms.
9. After mitochondrial genome assembly, the authors are mentioning about the read trimming, which is wrong.
10. Annotation was done using two different servers viz., MITOS and MITOS2? How have the authors made a consensus of the two outputs?
11. Please use the terms “genome assembly” and “mitochondrial genome assembly” to avoid any confusion. The methodology section needs to be revised thoroughly and I suggest the authors provide a flowchart of work.
Results and discussion
1. In results, the authors have mentioned the two different genome sizes based on two different k-mers. What is the exact genome size authors have found in their analysis? Authors must have streamlined the analysis and come up with the final result.
2. In the section “Repetitive elements in the nuclear genome of Tridacna crocea”, “first analysis” and “second analysis” terms are confusing. I suggest authors make a consensus and provide the final results. Additional analysis or steps can be given as the Supplementary information.
3. What is the size of the mitochondrial genome? How have authors concluded the almost completeness of the mitochondrial genome despite missing several tRNA genes? In a recent study by Cai et al. (2019), the existing mitogenome of T. crocea is found to be 19,157 bp in length, containing 13 protein-coding genes, 2 rRNA genes, 24 tRNA genes, and 1 non-coding control region. Missing several tRNA genes may be attributed to errors in analysis pipelines. Authors must cross-check before making such a claim of getting a nearly complete mitochondrial genome (in the conclusion section also). Please cite the following paper:
(Cai S, Mu W, Wang H, Chen J, Zhang H. Sequence and phylogenetic analysis of the mitochondrial genome of giant clam, Tridacna crocea (Tridacninae: Tridacna). Mitochondrial DNA Part B. 2019 Jan 2;4(1):1032-3.)
Conclusion
1. As per the aforementioned comments, please revise the conclusion section.
Minor comments
1. Authors should format in-text referencing style throughout the manuscript; provide either the reference number or the authors’ name and year.
2. Once the (expanded) term Tridacna crocea is mentioned, use its abbreviated form i.e. T. crocea.
3. Line 75-76, Change it into- This study forms part of a broad effort aimed at developing genomic resources of T. crocea and other giant clams.
4. Line 78-79, Point no. (iii) is missing, correct it.
5. Reference to the Galaxy platform is missing.

Author Response

Comment: Baeza et al. carried out a genomic analysis of the boring giant clam Tridacna crocea, a species of bivalve in the family Cardiidae, to explore its genome size, unique genome content, repetitive elements, nuclear RNA gene operon, and mitochondrial chromosome assembly. This study provides the genomics resource of an endangered species which is listed in the Convention on International Trade in Endangered Species (CITES) appendix II (https://cites.org/eng), and the International Union for Conservation of Nature (IUCN) Red List of Threatened Species (Version 2017-2, www.iucnredlist.org). I commend the authors for taking this initiative which is crucial for the scientific measures and subsequent policy making to preserve the flora and fauna of tropical and subtropical shallow sea coral reef environments.

Answer: Thanks for the positive view of our manuscript

I have the following comments/suggestions for the authors:

Introduction

  1. Comment: What is low-pass short read next generation sequencing? Why have authors opted for this technique instead of deep sequencing? Answer: Low pass is synonym of low-coverage. We have clarified this upon first mention in the manuscript.

  1. Comment: What is the genome coverage authors have obtained using low-pass short read next generation sequencing? Mention it in the manuscript. Answer: The coverage is aprox. 6x, we have added this information to the manuscript. Thank you.

Methodology

  1. Comment: Line 107-109: “This dataset was generated by [22] Tan et (2021) to assemble the mitochondrial genome of T. crocea and examine phylogenetic relationships among representatives of the same subfamily Tridacninae.” What is this dataset here? Answer: the raw data and where to find it are mentioned at the end of the manuscript following journal rules
  2. Comment: During preprocessing of the data, what is the minimum read length selected by the authors? After quality filtering, what was the %age of paired-end and singleton reads? Have authors removed the singleton reads before the analysis? Explain it. Answer: Considering that this is a brief report, this ancillary information is not presented. The others referee did not have a problem with this lack of information as well as the editor. However, if the referee and editor feel strongly about this issue, we can add this information.
  3. Comment: Authors have used two different genome-assembly strategies viz., SPAdes and CAP3, and found the discrepancies in the results. Rather, was it not useful to de novo assemble the reads first using SPAdes followed by the contig assembly using CAP3? Answer: We respectfully disagree with the referee and we think that he/her did not the manuscript with attention. CAP3 is used as part of the repeatexplorer pipeline to find repetitive elements and the use of CAP3 does not represent an attempt to assemble the entire genome. Please, remember, this is a genome survey sequencing paper, not a draft genome paper. Second, we use SPAdes to confirm that the operon was filtered out by Kraken2, again, not to assemble the entire genome. We have read this section again and allowed colleagues to read it as well, no one detected problems with the explanation. The same holds true with other referees.
  4. Comment: Authors have used the Kraken2-microbial-fatfree database to filter out the microbial reads? Then, how is it possible that Kraken2 filtered out the 45 rRNA DNA operon genes? Have authors tried changing the parameters of Kraken2 before concluding such results? Answer: We respectfully disagree with the referee. It is well known that Kraken2 does filter out the 45S operon because of similarities among eumetazoans. It is a very well conserved gene and because of that reason is commonly used in phylogenetic reconstruction (as a side). Even with different parameters, that happens. No issue here.
  5. Comment: Why have the authors aligned the nuclear ribosomal operons of crocea with those of Perna canaliculus and Perna virdis? Answer: Because it was the only entire operon available for any other species of bivalve.
  6. Comment: What is the “reference mtDNA provided by us”? Answer: WE have modified the sentence and now we are writing ‘provided by the user’ instead of ‘provided by us’

  1. Comment: Authors have assembled the mitochondrial genome using de novo strategies using Trinity and rnaSPAdes. “In parallel, the pipeline assembled the genome using a genome-guided strategy (in the software Trinity), mixed the totality of the assembled contigs, mapped these assembled contigs to the reference mitochondrial genome, and finally converted the contigs into a mitochondrial genome assembly ([39] Nachtigall et al. 2021).” What do you mean by “parallel” and “converted the contigs into a mitochondrial genome assembly ([39] Nachtigall et 2021).” What is this reference for? Authors must be cautious about the technical terms. Answer: We are not understanding the comment by the referee. We are providing details about the pipeline and we are citing the author of the pipeline!
  2. Comment: After mitochondrial genome assembly, the authors are mentioning about the read trimming, which is wrong. Answer: The referee is not correct. We already trimmed reads and QC them as the first step in our workflow, and as indicated in section 2.3.
  3. Comment: Annotation was done using two different servers , MITOS and MITOS2? How have the authors made a consensus of the two outputs? Answer: Thanks for the opportunity to clarify this point. Consensus was achieved while manually curating the two annotations, including start and stop codon error correction. We have provided details to the readers in this new version of the manuscript.
  4. Comment: Please use the terms “genome assembly” and “mitochondrial genome assembly” to avoid any confusion. The methodology section needs to be revised thoroughly and I suggest the authors provide a flowchart of work. Answer: We have double checked and used genome assembly and mitochondrial genome assembly throughout the manuscript. Because this is a brief report, we are not providing a flowchart. We have discussed the issue and we think the workflow is clearly explained in the manuscript.

Results and discussion

  1. Comment: In results, the authors have mentioned the two different genome sizes based on two different k-mers. What is the exact genome size authors have found in their analysis? Authors must have streamlined the analysis and come up with the final Answer: We respectfully disagree with the referee. It is well known that genome size estimation is a function of kmer size when coverage (read numbers) is constant. Some studies do estimate genome size using a single kmer size but that is incorrect. It is better to be transparent and explore the effect of kmer size as done here.
  2. Comment: In the section “Repetitive elements in the nuclear genome of Tridacna crocea”, “first analysis” and “second analysis” terms are confusing. I suggest authors make a consensus and provide the final results. Additional analysis or steps can be given as the Supplementary information. Answer: We respectfully disagree with the referee. We have talked this point with the referee and they appreciated the transparency with the two different analyses showing differences and the effect of read cleaning. The second referee and editor did have no problem with our approach.
  3. Comment: What is the size of the mitochondrial genome? How have authors concluded the almost completeness of the mitochondrial genome despite missing several tRNA genes? In a recent study by Cai et al. (2019), the existing mitogenome of crocea is found to be 19,157 bp in length, containing 13 protein-coding genes, 2 rRNA genes, 24 tRNA genes, and 1 non-coding control region. Missing several tRNA genes may be attributed to errors in analysis pipelines. Authors must cross-check before making such a claim of getting a nearly complete mitochondrial genome (in the conclusion section also). Please cite the following paper: (Cai S, Mu W, Wang H, Chen J, Zhang H. Sequence and phylogenetic analysis of the mitochondrial genome of giant clam, Tridacna crocea (Tridacninae: Tridacna). Mitochondrial DNA Part B. 2019 Jan 2;4(1):1032-3.) Answer: We considered the mitochondrial genome nearly complete based on the percentage retrieved from RNAseq reads that comprised 92.06% of the reference mitochondrial genome (reference length: 18,266 bp). This is clearly explained in the manuscript. Although there is no clear cut, mitogenomes assembled with more than 75% of the references are named nearly complete. We understand the point by the referee nonetheless, the manuscript indicated by the referees does not comply with FAIR recommendations/rules in data science. The paper mentioned by the referee does not have raw reads deposited in genbank and thus, we cannot check for the accuracy of that assembly. By contrast, we can check the accuracy of the assembly of the mitogenome we have used. Considering that the other referee did not have a problem with our approach, we decided not to use that second mitogenome (that we were aware of).
  4. Conclusion

  1. Comment: As per the aforementioned comments, please revise the conclusion Answer: Modified as requested by referee.

Minor comments

  1. Comment: Authors should format in-text referencing style throughout the manuscript; provide either the reference number or the authors’ name and year.

Answer: WE have done as requested. Thank you.

Comment: Once the (expanded) term Tridacna crocea is mentioned, use its abbreviated form i.e. T. crocea. Answer: Done, as requested by the referee. However, notice that when the species is starting a sentence, needs to be fully spelled to follow zoological rules. The name also needs to be complete in figure captions, table titles, and subtitles in the main text of the manuscript, in line with rules.

Comment: Line 75-76, Change it into- This study forms part of a broad effort aimed at developing genomic resources of T. crocea and other giant clams. Answer: We respectfully disagree with the referee, the use of ‘of’ in the sentence proposed is not grammatically correct.

Comment: Line 78-79, Point no. (iii) is missing, correct it. Answer: Corrected. Thank you.

Comment: Reference to the Galaxy platform is missing. Answer: Cited, as requested by the referee. Thank you.

Reviewer 2 Report

This manuscript provides valuable information regarding genomic survey and resources for the boring giant clam Tridacna crocea, particularly as it s an evolutionary, ecologically, economically, and culturally important reef-dwelling bivalve targeted by a profitable ornamental fishery in the Indo- Pacific Ocean. This information is essential to guide a chromosome-level assembly of the nuclear genome of T. crocea, and will be useful for future conservation or fishery management in this area. The MS has interesting topic and is nicely written. The results are clear. I have only few questions to the authors.

(1) Provide more details about the specimen sampled (e.g. length, weight). 

(2) I recommend upload mitochondrial sequences assembled from RNAseq to NCBI Genbank. 

Author Response

Thank you for the positive view. We have uploaded the sequence to genbank and we are waiting for a genbank number. We will make sure to have it in time in the case the paper is accepted.

Round 2

Reviewer 1 Report

The authors have attempted to answer all the queries raised by me.

I still suggest authors add a methodology workflow chart as supplementary material for the brief report. It will be better for readers to get a clear-cut view of the methodology.

Besides, authors must adhere to the reference formatting of the journal.